# PROBABILITY OF MATCHING FOR PARETO COVERAGE

## ABSTRACT

In batch multi-objective Bayesian optimization (MOBO), it is often desirable to identify the whole Pareto optimal set, especially when considering the complicated interplay between different design criteria and constraints. This poses unique challenges in acquiring batches of both high quality and diversity to cover the Pareto front. We propose a novel acquisition strategy, Probability of Matching, which evaluates both batch candidate quality and diversity by explicitly capturing the likelihood that a batch matches the true Pareto set. This is achieved by factorizing the probability into two components: the likelihood that all batch points are Pareto optimal, and the probability that they collectively cover the full Pareto set. To estimate the coverage probability and promote diversity, we incorporate space-filling design principles, resulting in our space-filling qEHVI (qEHVI-SF), a new batch MOBO method. Across synthetic benchmarks and real-world tasks, qEHVI-SF consistently outperforms state-of-the-art baselines on standard MOBO metrics as well as a new design-space coverage metric, Expected Minimum Distance (EMD), with comparable computational efficiency.

## 1 INTRODUCTION

Bayesian optimization (BO) is a powerful framework for optimizing unknown or uncertain objective functions that are complex (non-linear, non-convex, non-stationary) and expensive to evaluate in terms of both time and cost (Tu et al., 2022; Garnett, 2023; Ahmadianshalchi et al., 2024; Wang et al., 2024). BO operates by sequentially querying objective function values for selective input, due to the "black-box" nature of the problem setting. To achieve the desired sampling and computation efficiency, it models the objective function probabilistically, constructs an acquisition function considering model uncertainty, and uses this acquisition function to iteratively select new points in the design space to query. The probabilistic model is updated with new observations at each iteration.

When multiple objectives are involved, BO becomes more complex compared to those targeting single-objective problems. Instead of identifying a single global optimum, the goal here is to approximate the non-dominated Pareto optimal set. In this context of multi-objective BO (MOBO), acquisition functions must evaluate the quality of samples relative to the unknown Pareto front. A widely used approach is by the Expected Hypervolume Improvement (EHVI; Emmerich et al. (2006)), which measures improvement in the hypervolume dominated with respect to a reference point. However, the effectiveness of EHVI is sensitive to the choice of the reference point, which can bias sampling toward specific regions of the objective space (Auger et al., 2009).

Batch Bayesian optimization can partially mitigate diversity limitations by suggesting multiple candidates simultaneously, reducing the risk of oversampling in a single localized region. However, batch selection in MOBO is inherently more challenging due to several factors: the acquisition function must account for multiple objectives; it must capture interactions among batch points to promote diversity and reduce redundancy; and it must carefully balance exploration of the design space with exploitation of known high-performing regions.

In this paper, we propose a novel acquisition function from a probabilistic angle: Probability of Matching, which can guide batch MOBO to better balance the trade-off between exploiting regions with known high performance and ensuring diversity among batch points. This new acquisition function captures the likelihood that an acquired batch matches the true Pareto optimal solutions as much as possible. To estimate this Probability of Matching effectively, we factorize it as a joint probability with two components: (i) the probability that all batch points belong to the Pareto set,

and (ii) the probability that the batch collectively covers all Pareto optimal solutions. By jointly estimating these two components during batch MOBO, our approach dynamically balances query candidate quality with solution diversity, enabling more effective multi-objective batch acquisition.

The quality of batch candidates is estimated by parallel Expected Hypervolume Improvement (qE-HVI; Daulton et al. (2020)), a batch version of EHVI. For the diversity within each batch, we incorporate principles from space-filling design, a well-established strategy in batch experimental design (Pronzato & Müller, 2012). Classical space-filling approaches, such as the minimax-distance design (Johnson et al., 1990), aim to uniformly distribute samples across the design space. Building on these estimates, our batch MOBO acquisition function formulation based on the Probability of Matching leads to new space-filling qEHVI (qEHVI-SF), which selects query points to maximize the minimum distance between Pareto optimal solutions and their nearest previously sampled points, thereby encouraging broader and more uniform coverage of the optimal region.

To validate the effectiveness of our Probability of Matching framework and space-filling estimation strategy, we evaluate our qEHVI-SF method across a range of challenging multi-objective optimization tasks. Compared to baseline methods, including qEHVI and the diversity-aware Quantile Stein Variational Gradient Descent (QSVGD; Gong et al. (2019)), our qEHVI-SF consistently achieves superior rediscovery performance across standard evaluation metrics, including hypervolume, inverted generational distance (IGD; Ishibuchi et al. (2015)) and other widely used benchmarks for MOBO. We also introduce a new metric, Expected Minimum Distance (EMD), to evaluate coverage in the design space, where our method again outperforms existing baselines. In addition, for the materials discovery case study, qEHVI-SF demonstrates superior performance with consistently better rediscovery ratio. Notably, these improvements come with minimal additional computational overhead relative to qEHVI, highlighting the efficiency and general applicability of our approach.

## 2 Multi-objective Bayesian optimization

### 2.1 Background

Multi-objective Bayesian optimization (MOBO) can be formulated by considering optimizing an array of multiple objective functions with respect to the decision or design variables in a well-defined design space, $\boldsymbol{x} \in \mathcal{X}$:

$$\boldsymbol{f}(\boldsymbol{x}) = (f_1(\boldsymbol{x}), f_2(\boldsymbol{x}), \ldots, f_m(\boldsymbol{x})), \qquad (1)$$

where $f_1(\cdot), \ldots, f_m(\cdot)$ are multiple design objectives defined similarly as in single-objective Bayesian optimization (SOBO). Unlike SOBO, which aims to identify a single global or local optimum, the objective of MOBO is to generate a set of designs that capture the best trade-offs across competing objectives, forming the *Pareto front*. Denote the image of the $m$ objective function as $\mathcal{Y} \subset \mathbb{R}^m$, the Pareto front is defined as $\mathcal{Y}^* = \{\boldsymbol{y} \in \mathcal{Y} : \nexists \boldsymbol{y}' \text{ s.t. } \boldsymbol{y}' \prec \boldsymbol{y}\}$ if the goal is to minimize each objective. Here, $\boldsymbol{y} \prec \boldsymbol{y}'$ reads as $\boldsymbol{y}$ *dominates* $\boldsymbol{y}'$ in the context of minimization for all $m$ design objectives, meaning that $\forall i \leq m, y_i \leq y_i'$ and $\exists j \leq m, y_j < y_j'$. In the design space, the pre-image of $\mathcal{Y}^*$ is denoted by $\mathcal{X}^*$, called the *Pareto optimal set*.

Batch MOBO extends the standard sequential MOBO by selecting a set of $q > 1$ query design points per iteration, enabling parallel evaluation of these multiple expensive black-box objectives. The goal is still to approximate the Pareto optimal set $\mathcal{X}^*$, but evaluations are performed in batches of size $q$:

$$\boldsymbol{X} = \left\{ \boldsymbol{x}^{(1)}, \ldots, \boldsymbol{x}^{(q)} \right\} \subseteq \mathcal{X}. \qquad (2)$$

At each iteration $n$, typical implementations of batch MOBO select the batch $\boldsymbol{X}_{n+1}$ by jointly maximizing a batch acquisition function $\alpha(\boldsymbol{X} \mid \mathcal{D}_n)$, where $\mathcal{D}_n = \{(\boldsymbol{x}_i, \boldsymbol{f}(\boldsymbol{x}_i))\}_{i=1}^n$ is the set of observed data points. The batch selection rule is given by:

$$\boldsymbol{X}_{n+1} = \arg \max_{\boldsymbol{X} \subseteq \mathcal{X}, |\boldsymbol{X}|=q} \alpha(\boldsymbol{X} \mid \mathcal{D}_n). \qquad (3)$$

One widely adopted batch acquisition function in MOBO is qEHVI, which generalizes EHVI to the batch setting by jointly approximating the hypervolume improvement of the entire batch (Emmerich et al., 2006). Assuming $A \subset \mathcal{Y}$ is a finite set of objective vectors, the hypervolume indicator $\mathcal{H}(A)$ is defined as the Lebesgue measure of the region in the objective space dominated by at least one

element of $A$ and bounded above by a reference point $\boldsymbol{r} \in \mathbb{R}^m$, which is dominated by all elements in the Pareto front:

$$\mathcal{H}(A) = \mathrm{Vol}\left(\{\boldsymbol{y} \in \mathbb{R}^m | \boldsymbol{y} \prec \boldsymbol{r} \text{ and } \exists\, a \in A : a \prec \boldsymbol{y}\}\right), \tag{4}$$

where $\mathrm{Vol}(\cdot)$ quantifies the size of a set of the $m$-dimensional objective space.

Given a batch of $q$ input points $\boldsymbol{X}$, the qEHVI acquisition function evaluates the expected increase in the dominated hypervolume over the current Pareto front set $\mathcal{Y}_n^*$:

$$\alpha_n^{\mathrm{qEHVI}}(\boldsymbol{X} \mid \mathcal{D}_n) := \mathbb{E}_{\boldsymbol{y}^{(1:q)} \mid \mathcal{D}_n, \boldsymbol{X}}\left[\mathcal{H}(\mathcal{Y}_n^* \cup \boldsymbol{y}^{(1:q)}) - \mathcal{H}(\mathcal{Y}_n^*)\right], \tag{5}$$

where $\boldsymbol{y}^{(1:q)}$ is the (random) vector of objective values for points $\boldsymbol{x}^{(1:q)}$ under the Gaussian process (GP; Rasmussen (2003)) surrogate model, and the expectation is taken over the joint posterior distribution of $\mathbf{f}(\boldsymbol{X})$ conditioned on current observations $\mathcal{D}_n$.

In practice, this expectation is often approximated using Monte Carlo integration by sampling $\boldsymbol{y}^{(1:q)}$ from the posterior and computing the average hypervolume improvement. In MOBO, qEHVI can be biased towards certain regions, depending on the shape of the Pareto front and the position of the reference point (Auger et al., 2009). For example, if the reference point is far from the Pareto front, qEHVI may prioritize extreme solutions on the Pareto front (Tian et al., 2016). This bias occurs because extreme solutions are likely to yield the greatest hypervolume improvement. However, the ultimate goal of MOBO is to collect evaluation samples across the entire Pareto front to support more informed design decisions. This calls for new decision-making strategies that ensure better coverage performance during acquisition.

## 2.2 RELATED WORK

Several existing methods have been developed to improve MOBO performance through better coverage of the Pareto front, but most of these operate directly in the objective space. For example, Expected MaxiMin Improvement (EMMI; Olofsson et al. (2018)) enhances Pareto front coverage by prioritizing improvement in unexplored regions of the objective space. Similarly, Inverted Generational Distance with Noncontributing Solutions (IGD-NS; Tian et al. (2016)) aims to improve coverage by directly minimizing the IGD metric (Zhou et al., 2006). These methods are effective when Pareto optimal solutions are densely clustered in a specific region in the design space. However, these approaches struggle to capture the full Pareto front when the solutions are widely dispersed across the design space. Implementing diversity improvement directly in the objective space also presents several challenges: 1) Validity: It is unclear whether any candidates actually exist in the regions of the objective space being targeted; 2) Bias: The estimation of these acquisition function values depends on GP surrogate modeling. This reliance introduces the risk of bias, similar to the challenge faced by EHVI, where the indicator may favor specific regions based on initialization. 3) Misalignment with optimization goals: Diversity improvement strategies in the objective space may suggest queries that may not align with the same preferences considering hypervolume improvement, potentially reducing the overall optimization performance.

Given the limitations of promoting diversity in the objective space, enhancing the diversity of Pareto optimal solutions in the design space presents a more reliable alternative. This is because: 1) the feasible region is explicitly defined, eliminating concerns about solution validity; 2) diversity is evaluated over the design space, making diversity modeling independent of potential biases in objective estimation; 3) promoting diversity in the design space does not compromise solution quality, as there is no inherent preferential direction within the feasible domain; 4) and moreover, diversity estimation in the design space does not introduce additional sensitivity to observational noise, if the diversity estimation is computed purely from the design space distribution of the candidates, thereby preserving the robustness of the approach.

However, not many related works have taken into account the diversity of Pareto optimal solutions when developing MOBO. Here, before we present our new batch MOBO strategy, we first consider a modified strategy, following QSVGD (Gong et al., 2019), which has been developed to increase the exploration ability in single-objective BO. We extend the original implementation into batch MOBO and still refer to it as QSVGD throughout the paper. QSVGD incorporates batch entropy to enhance the diversity of queried samples. This formulation, though effective for global exploration, does not directly target improved coverage of the Pareto front. Therefore, here we consider this

multi-objective QSVGD as a baseline when evaluating our direct coverage improvement strategy detailed later. The acquisition function for QSVGD is defined as follows:

$$\alpha_n^{\text{QSVGD}}(\boldsymbol{X} \mid \mathcal{D}_n) := \mathbb{E}_{\boldsymbol{y}^{(1:q)} \mid \mathcal{D}_n, \boldsymbol{X}} \left[ \mathcal{H}(\mathcal{Y}_n^* \cup \boldsymbol{y}^{(1:q)}) - \mathcal{H}(\mathcal{Y}_n^*) \right] + \eta \mathscr{H}(\boldsymbol{X}), \tag{6}$$

where $\mathscr{H}$ is the entropy of the distribution over the batch samples and $\eta$ is the hyperparameter to balance the batch quality and diversity.

## 3 BATCH MOBO WITH SPACE FILLING

We now describe our new batch MOBO method based on the key concept of Probability of Matching to guide queried batches to approach and cover the Pareto front. With a space-filling strategy to efficiently approximate the coverage probability, the corresponding batch MOBO iterations can better balance the trade-off between exploiting regions with known high performance and ensuring diversity among batch points.

### 3.1 PROBABILISTIC PARETO SET MATCHING

We propose a novel batch acquisition strategy for MOBO that optimizes a new metric, the Probability of Matching, to guide batch selection. This metric measures the probability that the selected batch is identical to the true Pareto optimal set by quantifying the likelihood that an acquired batch not only contains high-quality solutions, but also closely approximates the entire Pareto optimal set.

Unlike existing approaches such as QSVGD that combine separate quality and diversity terms through additive objectives, our method models both aspects jointly within a single probabilistic framework. This removes the need for sensitive hyperparameter tuning, which is often required due to the varying characteristics of different black-box objectives and can be both computationally costly and difficult to calibrate during the optimization of new problems. More importantly, our approach avoids fallback mechanisms that only adjust the trade-off when a failure is detected or performance begins to degrade. By relying on a single coherent metric from the outset, the acquisition strategy proactively maintains a high level of search efficiency and robustness, reducing the risk of oversampling in extreme value regions or inefficient exploration.

To properly estimate the intractable Probability of Matching, we decompose it into two interpretable components:(1) the probability that all points in the batch are Pareto optimal, and (2) the probability that the batch collectively covers the full Pareto set. By estimating these two components jointly during the acquisition process, our method maintains a dynamic balance between candidate quality and diversity. This unified perspective enables the acquisition function to favor batches that not only yield good performance but also provide broader coverage of the Pareto optimal set.

We model and optimize the probability that the acquired batch $\boldsymbol{X}$ approximates the true Pareto optimal set $\mathcal{X}^*$ both in quality and distributional diversity. Specifically, we explicitly consider the probability that the acquired batch $\boldsymbol{X}$ approaches $\mathcal{X}^*$:

$$P(\boldsymbol{X} = \mathcal{X}^*) = P(\mathcal{X}^* \subseteq \boldsymbol{X}, \boldsymbol{X} \subseteq \mathcal{X}^*) = P(\boldsymbol{X} \subseteq \mathcal{X}^*)P(\mathcal{X}^* \subseteq \boldsymbol{X} \mid \boldsymbol{X} \subseteq \mathcal{X}^*), \tag{7}$$

which quantifies the likelihood that the selected batch $\boldsymbol{X}$ matches exactly the sampled set of true Pareto optimal solutions. As in (7), we factorize this "matching" event $\boldsymbol{X} = \mathcal{X}^*$ by estimating the joint probability of $\mathcal{X}^* \subseteq \boldsymbol{X}$ and $\boldsymbol{X} \subseteq \mathcal{X}^*$: $\boldsymbol{X} \subseteq \mathcal{X}^*$ is the event of $\boldsymbol{X}$ belonging to the Pareto optimal set and $\mathcal{X}^* \subseteq \boldsymbol{X}$ is the event that all optimal solutions are contained within $\boldsymbol{X}$. This provides a more balanced criterion for batch selection in MOBO. Many optimal quality preferred methods, such as qEHVI, prioritize $\boldsymbol{X} \subseteq \mathcal{X}^*$ while neglecting $\mathcal{X}^* \subseteq \boldsymbol{X}$. This imbalance may explain qEHVI's tendency to favor extreme regions, where $\boldsymbol{X} \subseteq \mathcal{X}^*$ is easier to satisfy since those solutions are less likely to be dominated. To improve coverage performance, new strategies should explicitly account for the probability of $\mathcal{X}^* \subseteq \boldsymbol{X}$ to promote coverage or diversity.

Regarding the coverage of the Pareto optimal set, batch MOBO is generally more effective than sequential counterparts, as a larger batch size naturally leads to higher $P(\mathcal{X}^* \subseteq \boldsymbol{X})$. On the other hand, excessively large batches may reduce $P(\boldsymbol{X} \subseteq \mathcal{X}^*)$. Also, such large batch sizes incur substantial costs due to the evaluation of numerous non-Pareto optimal solutions, resulting in inefficient utilization of the evaluation budget. Therefore, simply increasing the batch size should not be viewed

as a universal solution for improving coverage. Another drawback of using a large batch size is the increased computational cost (Binois et al., 2025). As the batch size $q$ grows, the number of possible batch combinations increases superlinearly, leading to significantly higher complexity in acquisition optimization and evaluation. Instead, we need a principled methodology that balances $P(\mathcal{X}^* \subseteq \boldsymbol{X})$ and $P(\boldsymbol{X} \subseteq \mathcal{X}^*)$ to achieve effective and diverse sample queries.

## 3.2 SPACE FILLING

To optimize (7), we first use normalized qEHVI to approximate $P(\boldsymbol{X} \subseteq \mathcal{X}^*)$ for batch $\boldsymbol{X}$. Next, we estimate the coverage probability $P(\mathcal{X}^* \subseteq \boldsymbol{X} \mid \boldsymbol{X} \subseteq \mathcal{X}^*)$, taking a space filling strategy (Pronzato & Müller, 2012). Since $\mathcal{X}^*$ denotes a continuous space while $\boldsymbol{X}$ is a finite set of sampled points, we evaluate the coverage capacity of $\boldsymbol{X}$ by defining $A_{\boldsymbol{X}}^r$ as the union of closed balls of radius $r$ centered at each point in $\boldsymbol{X}$. We then use $P(\mathcal{X}^* \subseteq A_{\boldsymbol{X}}^r \mid \boldsymbol{X} \subseteq \mathcal{X}^*)$ as a surrogate to estimate $P(\mathcal{X}^* \subseteq \boldsymbol{X} \mid \boldsymbol{X} \subseteq \mathcal{X}^*)$.

Note that the coverage probability is already conditioned on $\boldsymbol{X} \subseteq \mathcal{X}^*$, which implies that expanding the covered region $A_{\boldsymbol{X}}^r$ leads to improved coverage of the Pareto optimal set. Therefore, a natural thought is to maximize the total volume of $A_{\boldsymbol{X}}^r$. Given that the batch size of $\boldsymbol{X}$ and the radius $r$ are fixed, the sum of the volumes of the individual balls is also fixed. Thus, to increase the total covered space, it is beneficial to reduce the overlap between balls. We achieve this by maximizing the minimum distance among points in $\boldsymbol{X}$, thereby encouraging a more dispersed and space-filling configuration. Besides the minimum distance term between the candidates inside the batch, we also take the previous observation location into account to ensure that the next-step acquisition does not overlap with the previously queried regions. The final acquisition function is defined as:

$$\mathbb{E}_{\boldsymbol{y}^{(1:q)}|\mathcal{D}_n, \boldsymbol{X}} \left[ \left( \mathcal{H}(\mathcal{Y}_n^* \cup \boldsymbol{y}^{(1:q)}) - \mathcal{H}(\mathcal{Y}_n^*) \right) \cdot \min \left\{ \Delta(\boldsymbol{X}, \boldsymbol{X}), \Delta(\boldsymbol{X}, \boldsymbol{X}_n) \right\} \right], \tag{8}$$

where $\Delta(\cdot, \cdot)$ denotes the minimum non-zero $L_2$ distance between two input sets.

It is crucial to emphasize that we do not recommend estimating coverage using the current Pareto optimal set $\mathcal{X}_n^*$. The definition of coverage probability is strictly conditioned on $\boldsymbol{X} \subseteq \mathcal{X}^*$. Substituting $\mathcal{X}_n^*$ for $\mathcal{X}^*$ implicitly changes the condition to $\boldsymbol{X} \subseteq \mathcal{X}_n^*$. This shift can lead to oversampling in the local region around $\mathcal{X}_n^*$ and undermine the goal of achieving comprehensive coverage over the true Pareto optimal set $\mathcal{X}^*$.

## 3.3 COMPLEXITY ANALYSIS

To demonstrate that our space-filling strategy does not introduce significant computational overhead compared to qEHVI, we present a complexity comparison of various batch MOBO acquisition strategies. For qEHVI, we have applied the box decomposition strategy to compute the expected hypervolume improvement, which has the time complexity of $\Theta(NmK(2^q - 1))$ (Daulton et al., 2020), since qEHVI requires computing the volume of $2^q - 1$ (the number of non-empty subsets of $\boldsymbol{y}^{(1:q)}$) hyper-rectangles for each of $K$ hyper-rectangles and $N$ Monte Carlo (MC) samples. The exact complexity of $K$ is unknown but it is super-polynomial in $m$ (Yang et al., 2019a). For QSVGD and qEHVI-SF, based on (7), the complexity consists of two parts: The first part is to estimate the probability of being Pareto optimal, which has the same complexity, $\Theta(NmK(2^q - 1))$, as similarly estimated in qEHVI. The second part is for approximating the coverage probability by either entropy or space filling. For QSVGD, we adapt the Kernel Density Estimation (KDE) to estimate the entropy in the design space, which has the complexity of $\Theta(q^2 d)$. For qEHVI-SF, we first estimate the minimum distance inside the batch, which has the complexity of $\Theta(q^2 d)$. Next, we estimate the minimum distance between the selected batch and previous evaluations, which has complexity of $\Theta(qnd)$. Thus, the overall complexity of coverage probability estimation for qEHVI-SF is $\Theta(q(q + n)d)$. For QSVGD and qEHVI-SF, the coverage probability estimation is performed in the design space and is independent of Pareto optimality estimation. Therefore, the overall complexity per iteration is $\Theta(NmK(2^q - 1) + q^2 d)$ for QSVGD, and $\Theta(NmK(2^q - 1) + q(n + q)d)$ for qEHVI-SF. At each iteration, the number of possible batch combinations of size $q$ from the candidate set $\mathcal{X}$ is $\binom{|\mathcal{X}|}{q}$, and we obtain $q$ evaluation candidates per iteration. Thus, the average time cost per evaluation is $\Theta\left(NmK\left(\frac{2^q - 1}{q}\right) + qd\right)\binom{|\mathcal{X}|}{q}$ for QSVGD, and $\Theta\left(NmK\left(\frac{2^q - 1}{q}\right) + (n + q)d\right)\binom{|\mathcal{X}|}{q}$ for qEHVI-SF. For qEHVI, the overall complexity per evaluation is $\Theta\left(NmK\left(\frac{2^q - 1}{q}\right)\right)\binom{|\mathcal{X}|}{q}$.

## 4 EMPIRICAL RESULTS

### 4.1 BASELINE BENCHMARK

We have performed experiments in different setups to investigate performance trends of different batch MOBO strategies with different batch sizes. To emphasize the importance of reliable coverage estimation of $P(\mathcal{X}^* \subseteq \boldsymbol{X})$, we focus on the MOBO problems that have multiple Pareto optimal regions in the corresponding design space. For the problems with only a single Pareto optimal region, optimization with respect to $P(\boldsymbol{X} \subseteq \mathcal{X}^*)$ is often sufficient, since if $\boldsymbol{X}$ does not belong to the current Pareto optimal region, it can be ignored as it can not be in any other Pareto optimal region.

We evaluate the performance of different acquisition strategies on two benchmark problems characterized by complex Pareto optimal solution distributions. The first is a simulated bi-objective Gaussian mixture (GM; Fröhlich et al. (2020)) function with a 2-dimensional design space, where the Pareto optimal solutions are widely dispersed. The second is the car side-impact problem (RE4-7-1), which involves a 7-dimensional design space and four objectives, with an unknown Pareto optimal set (Tanabe & Ishibuchi, 2020). Both experiments are implemented within the `BOTorch` framework (Balandat et al., 2020). Additional implementation details are provided in Appendix A.1.

For each problem, we evaluate optimization performance using two metrics. The first is hypervolume, which measures the quality of the solutions obtained in the objective space. The second is expected minimum distance (EMD), defined in (9), which quantifies the coverage of the Pareto optimal solutions in the design space. EMD is conceptually similar to the IGD, but it operates in the design space rather than the objective space. In MOBO, fully covering the Pareto front does not guarantee that all Pareto optimal designs are recovered. However, capturing all Pareto optimal designs does imply full coverage of the Pareto front. Therefore, EMD serves as a stricter metric than IGD, offering higher selectivity and precision in evaluating MOBO solution quality:

$$\text{EMD}(\boldsymbol{X}_n, \mathcal{X}^*) = \frac{1}{|\mathcal{X}^*|} \sum_{\boldsymbol{x}^* \in \mathcal{X}^*} \min_{\boldsymbol{x} \in \boldsymbol{X}_n} \|\boldsymbol{x} - \boldsymbol{x}^*\|_2. \tag{9}$$

As shown in Figure 1, qEHVI-SF consistently outperforms competing baselines in both candidate quality (measured by hypervolume) and diversity (measured by EMD) in both problems. Moreover, its performance remains stable across different batch sizes. In contrast, qEHVI and QSVGD exhibit high sensitivity to batch size. For example, on the first GM problem, qEHVI performs the best with a batch size of two; while for RE4-7-1, it performs the best with a batch size of ten. QSVGD also shows significant variability depending on the batch size. These results indicate that qEHVI-SF is a more robust strategy under varying batch size settings. Additionally, when comparing results within the same column (i.e., at fixed batch sizes), qEHVI-SF consistently demonstrates superior performance on both metrics, particularly when the batch size is small. Also, results by qEHVI-SF have smaller standard deviation values across trials, indicating that qEHVI-SF is a more robust acquisition strategy with different random initialization throughout optimization iterations than other strategies.

Additional results on standard MOBO benchmarks such as ZDT (Zitzler et al., 2000) and DTLZ (Deb et al., 2005) families are provided in Appendix A.2.

### 4.2 REAL-WORLD CASE STUDIES

We further evaluate and compare batch MOBO performance in an alloy inverse design task aimed at identifying compositions with targeted material properties. Specifically, we aim to recover Pareto optimal compositions from a pool of 1,000 given candidates under a constrained evaluation budget (Hastings et al., 2024). To provide a comprehensive assessment on the effectiveness of different batch MOBO strategies, we vary the batch size and measure the resulting Pareto optimal solution rediscovery ratio, for qEHVI, QSVGD, and our qEHVI-SF.

The design objectives in this MOBO problem include six material properties: Stacking Fault Energy (SFE), principal axial elastic constant ($C_{11}$), Heat Capacity (HC), Thermal Conductivity (TC), Solidification Range (SR), and Room-Temperature Density (RTD). As a model of the trade-offs inherent to alloy design, these six material properties—SFE, $C_{11}$, HC, TC, SR, and RTD—collectively

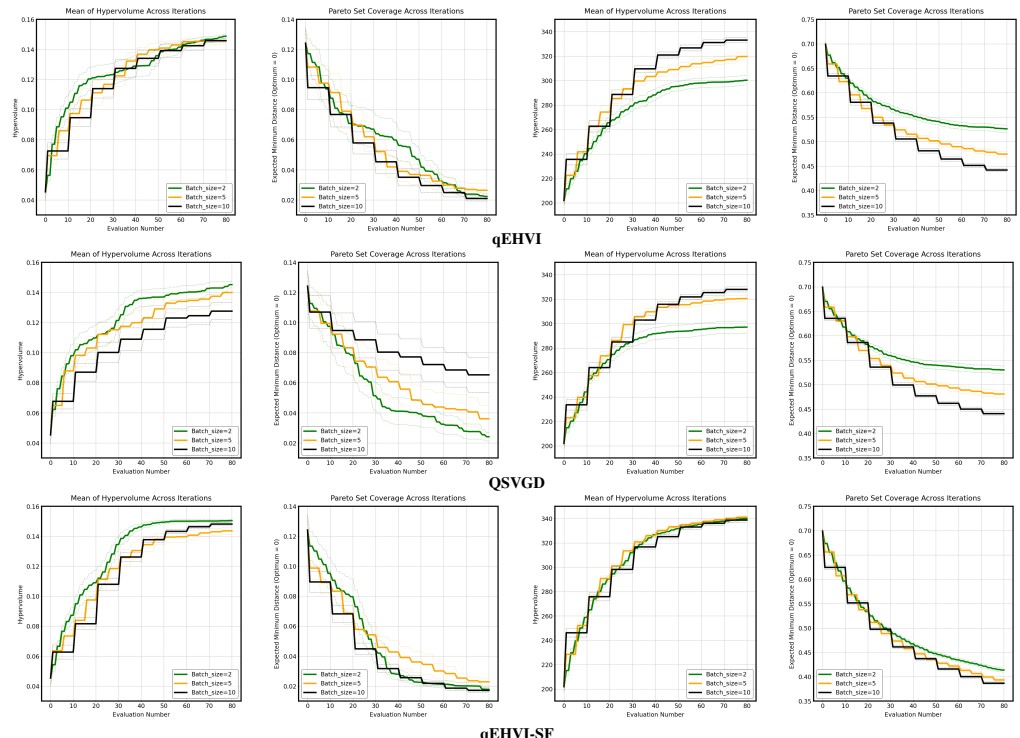

Figure 1: Comparison of batch MOBO performances under different experimental settings across three acquisition strategies: qEHVI, QSVGD, qEHVI-SF. The first two columns show the changes of hypervolume and expected minimum distance (EMD) across MOBO iterations for the GM problem, and the last two columns correspond to the results for the RE4-7-1 problem.

define the key thermodynamic, mechanical, and processing constraints that shape performance. These attributes are deeply entangled: they do not vary independently, and optimizing one often perturbs others. Their interplay reflects the practical complexity of real-world alloy development, where performance gains are rarely free and improvements in one dimension often impose costs elsewhere.

SFE governs dislocation behavior, including splitting, cross-slip, and twinning, and directly influences deformation mechanisms such as transformation-induced plasticity (TRIP) and twinning-induced plasticity (TWIP) (Gallagher, 1970; Ahlers, 1970; Vitos et al., 2006; Pierce et al., 2015). $C_{11}$ characterizes the intrinsic stiffness along the primary crystallographic axes and plays a key role in determining both the yield strength and elastic anisotropy (Hill, 1948; Yoo, 1986; Hutchinson, 2015). Thermal properties fall into two regimes: HC, which governs transient thermal response by storing or releasing energy; and TC, which dictates steady-state heat transport by controlling spatial heat diffusion (Callister & Rethwisch, 2013). Both properties are critical in thermally demanding systems, from aerospace components to microelectronic devices (Hanuska et al., 2000; Zawada, 2006). These thermal properties often exhibit an inverse relationship with RTD, a measure of mass per unit volume. While lower density can enhance specific properties, it may compromise absolute stiffness, thermal conductivity, or mechanical robustness (Ashby & Cebon, 2005; Callister & Rethwisch, 2013). SR reflects the temperature interval over which an alloy transitions from liquid to solid. A narrow SR minimizes elemental segregation and reduces the risk of hot cracking during solidification, thereby enhancing processability. In contrast, a wider SR expands the accessible compositional design space but increases susceptibility to solidification defects (Flemings, 1974).

Simultaneously optimizing all six dimensions is inherently constrained by competing material mechanisms. For example, alloying strategies to increase ductility via reduced SFE or enhance stiffness via increased $C_{11}$ may inadvertently raise density or broaden SR, thus worsening processability or reducing mass efficiency (Ashby, 2011c;a;b). These trade-offs cannot be resolved independently; rather, they manifest as interlocked constraints in multi-objective optimization landscapes. The alloy design process must therefore navigate a multi-dimensional Pareto front, where gains in one domain

often exact cost in another, a reality mirrored in recent multi-objective Bayesian optimization frameworks that explicitly balance candidate quality with design space coverage (Khatamsaz et al., 2021; 2022).

Here, we study six constructed MOBO tasks by grouping six properties into three bi-objective (Bi-1: $C_{11}$ and SFE , Bi-2: HC and TC, Bi-3: SR and RTD), two tri-objective (Tri-1: RTD, $C_{11}$, and SFE, Tri-2: SR, HC, and TC), and all six objective tasks (All: SR, RTD, HC, TC, $C_{11}$, and SFE). For each property, we first train a property predictor on the full candidate set and use this surrogate model as the black-box objective function. We initialize each batch MOBO run with 10 randomly selected compositions and allocate a total of 80 evaluations. The goal is to compare acquisition strategies in terms of how many Pareto optimal solutions they are able to recover within this evaluation budget. We vary the batch size from the set $2, 5, 10$, corresponding to $40, 16, 8$ batch MOBO iterations.

We use the following metrics to evaluate the re-identification performance: 1) Rediscovery ratio; 2) Hypervolume; 3) EMD; 4) IGD; 5) Maximum Spread (Zitzler & Thiele, 2002); and 6) Spacing (Schott, 1995). Rediscovery ratio is defined as the number of rediscovered Pareto optimal solutions divided by the total number of true Pareto optimal solutions. It serves as the most practical evaluation metric for materials inverse design, as it directly measures the effectiveness of recovering target compositions. Therefore, we adopt it as the primary metric for comparison. Figure 2 presents the rediscovery ratios achieved by different acquisition strategies across various experimental setups. The evaluation results for other metrics and the acquisition behavior analysis exhibit consistent trends with those observed in the benchmark examples (see Appendix A.3).

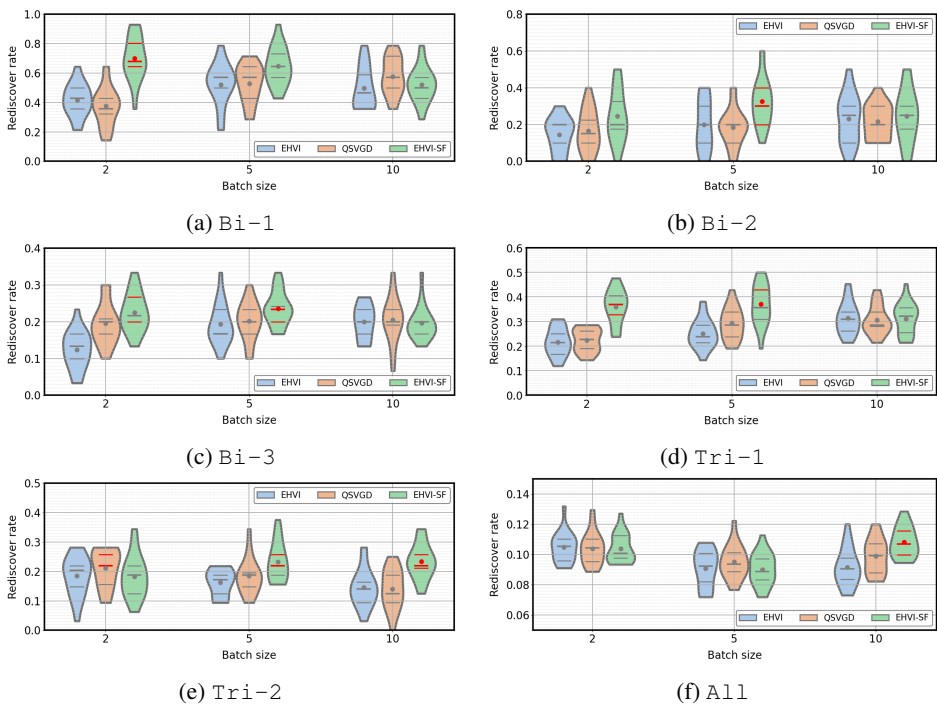

Figure 2: Rediscovery performance comparison for different MOBO settings: bi-objective combinations considering (a) Bi-1: $C_{11}$ and SFE, (b) Bi-2: HC and TC, (c) Bi-3: SR and RTD; tri-objective tasks considering (d) Tri-1: RTD, $C_{11}$, and SFE, (e) Tri-2: SR, HC, and TC; as well as considering all six objectives: (f) All: SR, RTD, HC, TC, $C_{11}$, and SFE. Dots: Mean value across 20 trials; Red: Best performance across all different settings.

For all the tasks, performing 80 random queries from 1000 candidates has a probability of 0.08 for any Pareto optimal solution being selected. Each method outperforms this baseline of 0.08. As the number of objectives increases, the rediscovery ratio decreases as expected, which is illustrated when comparing the trends in Figures 2a, 2d, and 2f; or similarly in Figures 2b, 2e, and 2f when increasing the number of objectives under consideration from two, three, to six.

Specifically, qEHVI-SF consistently rediscovers the most Pareto optimal solutions compared to qE-HVI and QSVGD across different settings. It typically shows stable performance with a batch size of five, although during the exploration stage (e.g., Figure 2f), this is not always the case. For qE-HVI, small batch sizes lead to low acquisition efficiency on tasks such as bi-objective ($C_{11}$ and SFE; HC and TC; SR and RTD) and tri-objective (RTD, $C_{11}$, and SFE) tasks. Performance improves with larger batch sizes, as the coverage probability $P(\mathcal{X}^* \subseteq \mathbf{X} \mid \mathbf{X} \subseteq \mathcal{X}^*)$ naturally increases. However, larger batches can reduce the probability $P(\mathbf{X} \subseteq \mathcal{X}^*)$, causing an overall performance drop as seen in Figures 2e and 2f.

For QSVGD, we employ a decaying schedule for the hyperparameter $\eta$ to gradually diminish the influence of the entropy term over iterations, thereby achieving a balance between exploitation and diversity (details in Appendix A.1). Finding the optimal exploration-exploitation balance remains challenging, as it varies by different scenarios. Without a dynamic balance, the diversity term may occasionally dominate the qEHVI term, causing the acquisition of candidates without any Pareto front potential. Consequently, QSVGD's overall performance is worse than qEHVI-SF and sometimes can even be worse than qEHVI.

### 4.3 COMPUTATIONAL EFFICIENCY

We validate the computational complexity derived in Section 3.3 by investigating the run-time of different batch MOBO strategies in our alloy design case study. Table 1 reports the average runtime required to generate one evaluation candidate across different acquisition strategies and experimental settings. The runtime is mainly influenced by the batch size $q$ and the number of objectives $m$. In general, incorporating coverage estimation introduces minimal additional overhead compared to qEHVI. For instance, with multiple objectives ($m = 6$), the cost is dominated by hypervolume estimation (Yang et al., 2019b), particularly for large $q$, since $(n + q)d \ll \frac{2^q - 1}{q}$. For qEHVI-SF, the coverage computation exhibits a higher standard deviation, likely due to the increase in computational complexity as the $n$ increases. In contrast, the other two strategies demonstrate more stable performance across trials. Meanwhile, the actual computational cost may deviate from the theoretical complexity analysis, as some acquisition optimization converges to the maximum before exhausting the optimization budget.

Table 1: Runtime (in seconds, Average±Standard Deviation) per candidate evaluation for different tasks, batch sizes, and BO strategies

| Acquisition strategies | Batch size | Bi-1 | Bi-2 | Bi-3 | Tri-1 | Tri-2 | All |
|---|---|---|---|---|---|---|---|
| qEHVI | 2 | 3.37±1.17 | 4.50±1.70 | 1.64±0.56 | 2.09±0.87 | 3.82±4.12 | 9.30±5.05 |
| | 5 | 4.72±2.35 | 2.44±1.30 | 2.28±0.75 | 3.79±2.01 | 1.77±0.63 | 46.03±52.18 |
| | 10 | 7.01±2.82 | 4.25±0.30 | 4.24±1.42 | 11.50±6.93 | 0.97±0.37 | 30.09±26.58 |
| QSVGD | 2 | 6.69±0.86 | 4.07±0.59 | 3.28±1.81 | 2.74±1.77 | 3.97±3.06 | 8.22±5.50 |
| | 5 | 3.88±1.30 | 7.32±1.62 | 2.35±1.55 | 3.97±1.71 | 1.80±0.54 | 56.23±57.17 |
| | 10 | 9.45±0.92 | 5.09±0.73 | 5.09±1.85 | 11.72±6.83 | 1.20±0.49 | 52.79±49.03 |
| qEHVI-SF | 2 | 5.12±0.91 | 3.57±4.15 | 2.38±1.14 | 3.11±2.33 | 8.86±6.08 | 10.07±8.68 |
| | 5 | 4.20±2.56 | 7.59±4.20 | 3.66±0.69 | 4.64±1.37 | 6.28±3.22 | 54.96±60.84 |
| | 10 | 7.58±2.73 | 3.79±1.64 | 6.48±0.65 | 13.23±5.63 | 6.89±4.18 | 52.01±70.60 |

## 5 CONCLUSION AND LIMITATION

In this study, we introduced qEHVI-SF, a novel batch acquisition strategy for batch MOBO, based on the concept of Probability of Matching. This approach emphasizes both the quality and diversity of batch candidates across optimization trials. Empirically, qEHVI-SF consistently demonstrates strong performance in producing high-quality solutions while maintaining better coverage of the Pareto front, all with only modest additional computational overhead compared to existing methods.

This work presents only one specific way of estimating the Probability of Matching, primarily through qEHVI to approximate probability of optimal and maximizing minimum distance to approximate coverage. While this surrogate proves effective in practice, the precise relationship between pairwise distance and true coverage probability remains unclear. Future work will focus on developing and evaluating more direct and theoretically grounded estimators for coverage probability. A deeper theoretical investigation into the relationship between distance-based heuristics and coverage probability is also necessary to further improve the robustness and interpretability of the proposed method.

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

# A  APPENDIX

## A.1  EXPERIMENTAL SETUP AND ENVIRONMENT

We consider the task of identifying Pareto optimal solutions from a discrete feasible set of 10,000 candidates for both the GM (Fröhlich et al., 2020) and RE4-7-1 (Tanabe & Ishibuchi, 2020) benchmark problems. All the evaluation experiments are implemented in the `BOTorch` framework (Balandat et al., 2020). Given a strict evaluation budget of 80 function queries per trial, each BO run is initialized with 10 randomly selected points. The corresponding GP surrogate models are trained using these initial evaluations. For each batch selection, we evaluate the acquisition function using MC sampling with 512 samples to ensure accurate estimation. The adopted optimization strategy for each iteration in `BOTorch` is set to be *joint* instead of *sequential* to further increase the estimation accuracy. Further details on the distinction between the *joint* and *sequential* batch selection strategies can be found in the prior work (Ament et al., 2023), which is adopted in the `BOTorch` framework.

For QSVGD, we assign equal weights to all quantiles, as none of our experimental setups incorporate a notion of risk. For the hyperparameter $\eta$, we define a piecewise schedule for the hyperparameter $\eta$: it is kept constant at $\eta_0$ for the first 40 iterations to maintain exploration, and then linearly decays to 0 over the next 40 iterations, effectively reducing the influence of the entropy term as optimization progresses. The choice of $\eta_0$ is task-dependent and is generally tuned so that the weighted entropy term is on a similar scale as the hypervolume improvement term, ensuring neither heavily dominates the optimization objective.

We compute the hypervolume using the default reference point, defined as 1.1 times the lower bounds of the objectives, for both the GM (Fröhlich et al., 2020) and RE4-7-1 (Tanabe & Ishibuchi, 2020) problems. The exact hypervolume is computed using the box decomposition method (Daulton et al., 2020). To compute EMD, we first exhaustively evaluate the objective values of all 10,000 candidates in the feasible space to obtain the true Pareto optimal solutions. We then compute EMD between the current non-dominated solutions and the true Pareto optimal solutions as a measure of convergence quality.

For all objective functions, the experimental runs were distributed across 8 Intel(R) Xeon(R) Gold 6248R CPUs.

## A.2  OTHER OBJECTIVE FUNCTIONS

We would like to note that in our experiments, we do not consider ZDT and DTLZ as our primary benchmarks due to the following observations: ZDT problems have Pareto solutions concentrated along the feasible space boundary, which tend to favor the exploitation-based methods including qEHVI, while DTLZ problems typically exhibit a single Pareto region in the design space, for which spatial diversity may not be needed to further improve batch MOBO performance. However, we here still provide the corresponding experimental results for these benchmark objective functions with detailed discussions.

Based on the results in Figure A.1, for the DTLZ2 problem, qEHVI-SF consistently outperforms other methods in terms of both hypervolume and EMD across all batch size configurations. However, since qEHVI already attains strong performance on this problem, the additional gains from the space filling strategy are relatively modest. This is likely because the Pareto optimal solutions for DTLZ2 are densely clustered within a single region, as Figure A.2 shows, limiting the opportunity for the enhanced coverage capabilities of qEHVI-SF to demonstrate its full advantage.

For the ZDT2 problem, qEHVI-SF consistently demonstrates more stable and competitive hypervolume performance across different batch sizes compared to other strategies. Meanwhile, its coverage metric improves as the batch size increases, suggesting enhanced diversity benefits in larger batches. However, similar to the DTLZ2 case, the baseline qEHVI already achieves strong performance due to the distribution of the Pareto optimal solutions lie along the boundary of the feasible space as Figure A.3. As a result, diversity oriented strategies like qEHVI-SF offer limited additional gains in such scenarios, since there is little room to further expand the coverage.

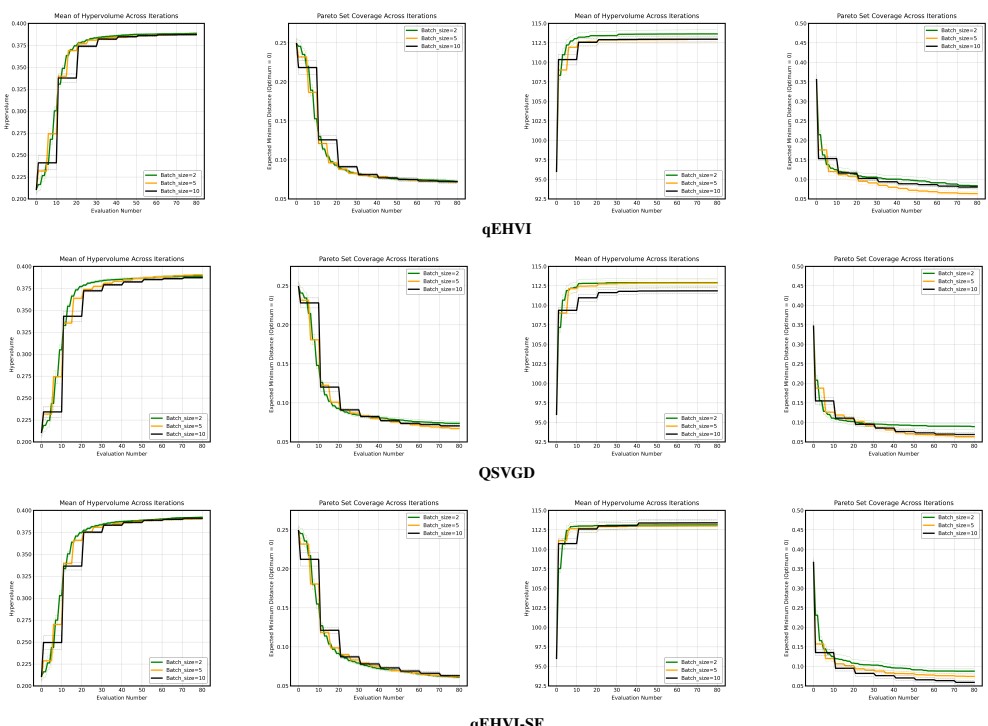

Figure A.1: Comparison of batch MOBO performances under different experimental settings across three acquisition strategies: qEHVI, QSVGD, qEHVI-SF. The first two columns show the changes of hypervolume and EMD across MOBO iterations for the DTLZ2 problem, and the last two columns correspond to the results for the ZDT2 problem.

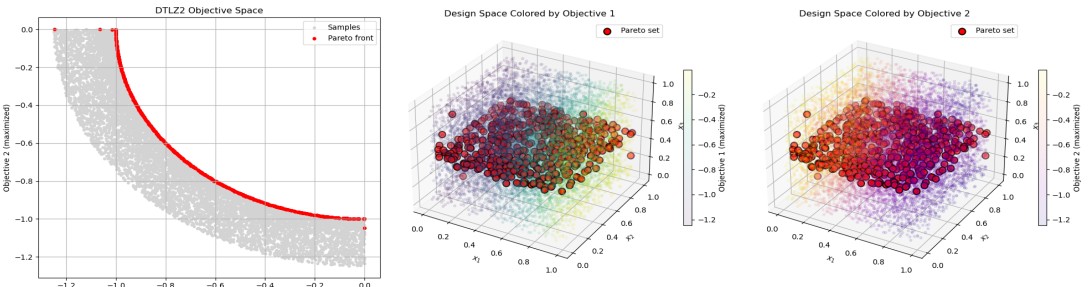

Figure A.2: Pareto optimal solution distribution of DTLZ2: The left plot shows the Pareto front in the objective space. The middle and right plots illustrate the distribution of Pareto optimal solutions in the design space, where each plot corresponds to one objective. The color bar indicates the respective objective function values.

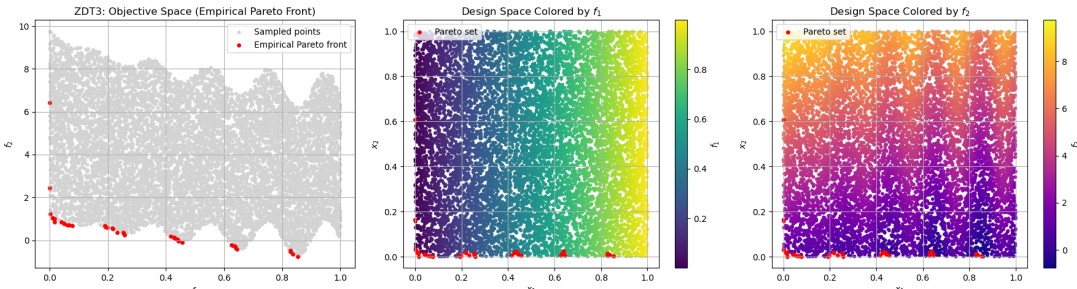

Figure A.3: Pareto optimal solution distribution of ZDT2: The left plot shows the Pareto front in the objective space. The middel and right plots illustrate the distribution of Pareto optimal solutions in the design space, where each plot corresponds to one objective. The color bar indicates the respective objective function values.

Due to the inherent characteristics of the DTLZ and ZDT problem classes where optimal solutions are either densely concentrated or confined to the boundary of the feasible space, these benchmarks may not effectively demonstrate the benefits of diversity enhancing strategies in MOBO.

### A.3    ADDITIONAL EVALUATION METRICS FOR ALLOY INVERSE DESIGN

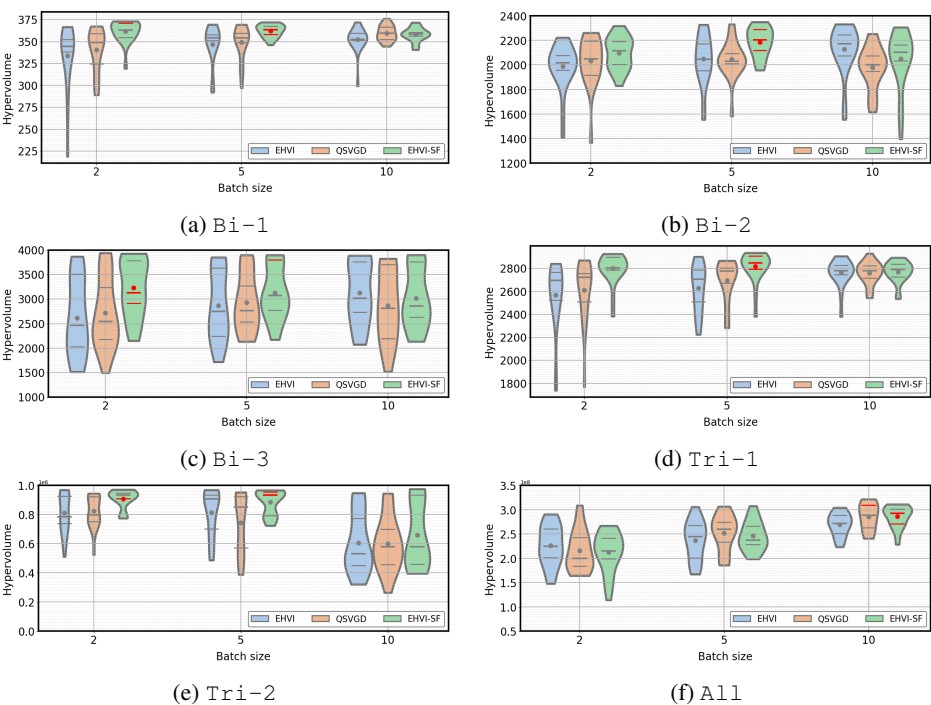

Figure A.4: Hypervolume comparison for different MOBO settings: (a) `Bi-1`; (b) `Bi-2`; (c) `Bi-3`; (d) `Tri-1`; (e) `Tri-2`; and (f) `All`. Dots: Mean value across 20 trials; Red: Best performance across all different settings.

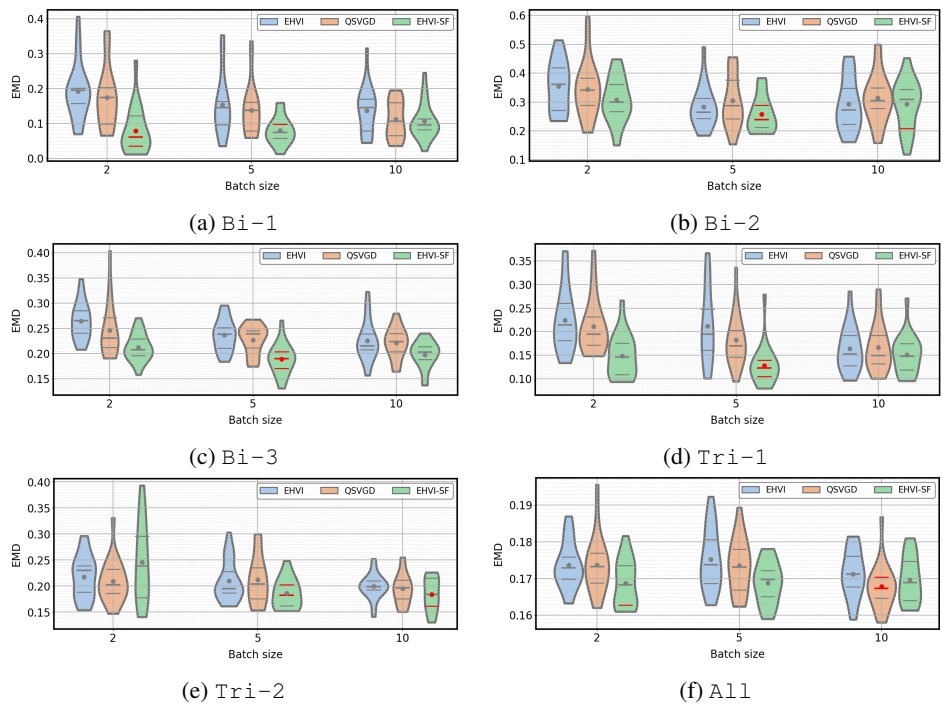

Figure A.5: Coverage comparison for different MOBO settings: (a) `Bi-1`; (b) `Bi-2`; (c) `Bi-3`; (d) `Tri-1`; (e) `Tri-2`; and (f) `All`.

In our materials inverse design tasks, in addition to the rediscovery ratio discussed in the main text, we also evaluate the performance of different acquisition strategies using several commonly adopted multi-objective evaluation metrics. These include Hypervolume, EMD, IGD (Ishibuchi et al., 2015), Maximum Spread (Zitzler & Thiele, 2002), and Spacing (Schott, 1995).

Here, we introduce three additional performance metrics in addition to the ones discussed in the main text: IGD, Maximum Spread (MS), and Spacing (SP). IGD assesses the coverage of the Pareto front in the objective space. However, since it operates in the target objective space and multiple Pareto optimal solutions can be mapped to the same objective values, IGD may be considered a weaker indicator of diversity compared to EMD. Maximum Spread measures whether the extreme value regions of the Pareto front are captured by the selected candidates. Spacing evaluates the uniformity of the batch distribution, reflecting how evenly the solutions are spread across the Pareto front. The definitions of these three additional evaluation metrics can be found in (10), (11), and (12):

$$\text{IGD}(\mathcal{Y}_n^*, \mathcal{Y}^*) = \frac{1}{|\mathcal{Y}^*|} \sum_{\boldsymbol{y} \in \mathcal{Y}^*} \min_{\boldsymbol{y'} \in \mathcal{Y}_n^*} \|\boldsymbol{y} - \boldsymbol{y'}\|_2, \tag{10}$$

$$\text{MS}(\mathcal{Y}_n^*) = \sqrt{\frac{1}{m} \sum_{k=1}^{m} \left( \max_{i=1,\ldots,|\mathcal{Y}_n^*|} y_k^{(i)} - \min_{i=1,\ldots,|\mathcal{Y}_n^*|} y_k^{(i)} \right)^2}, \tag{11}$$

$$\text{SP}(\mathcal{Y}_n^*) = \sqrt{\frac{1}{|\mathcal{Y}_n^*| - 1} \sum_{i=1}^{|\mathcal{Y}_n^*|} \left( d_i - \bar{d} \right)^2}, \tag{12}$$

where

$$\mathcal{Y}_n^* = \{\boldsymbol{y}^{(1)}, \boldsymbol{y}^{(2)}, \ldots, \boldsymbol{y}^{(|\mathcal{Y}_n^*|)}\}, \quad \boldsymbol{y}^{(i)} = (y_1^{(i)}, y_2^{(i)}, \ldots, y_m^{(i)}) \in \mathbb{R}^m$$

$$d_i = \min_{j=1,\ldots,|\mathcal{Y}_n^*|, j \neq i} \left\| \boldsymbol{y}^{(i)} - \boldsymbol{y}^{(j)} \right\|_1, \quad \bar{d} = \frac{1}{|\mathcal{Y}_n^*|} \sum_{i=1}^{|\mathcal{Y}_n^*|} d_i.$$

Figures A.4 and A.5 show the distribution of final performance across 20 trials for the hypervolume and EMD metrics, respectively. In terms of hypervolume, qEHVI-SF consistently outperforms all other acquisition strategies, which aligns well with the rediscovery ratio results reported in the main text. Furthermore, the narrow spread of hypervolume values across trials highlights the stability of qEHVI-SF relative to both qEHVI and QSVGD. For EMD, qEHVI-SF also demonstrates superior performance in most cases, except for the `all` composition scenario where it is slightly outperformed by QSVGD. However, despite QSVGD achieving marginally better coverage in this case, its performance in hypervolume and rediscovery remains suboptimal as Figures A.4f and 2f show. This suggests that QSVGD may sometimes overly emphasize diversity at the expense of solution quality, which is critical in tasks such as materials discovery where both diversity and objective quality are essential.

Table 2: Comparison of Pareto front coverage with respect to IGD↓

| Acquisition strategies | Batch size | Bi-1 | Bi-2 | Bi-3 | Tri-1 | Tri-2 | All |
|---|---|---|---|---|---|---|---|
| qEHVI | 2 | 38.74±30.78 | 4.21±2.37 | 18.60±11.63 | 30.77±22.01 | 19.71±7.00 | 45.11±2.43 |
|  | 5 | 28.84±14.99 | 3.51±2.05 | 14.55±8.50 | 23.65±10.68 | 20.21±4.87 | 45.46±2.41 |
|  | 10 | 26.23±10.77 | 2.92±2.17 | 10.07±5.88 | 17.57±6.51 | 25.23±8.48 | 43.62±1.45 |
| QSVGD | 2 | 37.41±31.06 | 3.86±2.56 | 15.83±9.34 | 27.49±18.60 | 18.94±3.70 | 45.57±2.56 |
|  | 5 | 25.40±13.71 | 3.49±1.75 | 12.23±5.88 | 21.17±8.53 | 23.23±7.34 | 44.43±2.40 |
|  | 10 | 19.54±10.14 | 4.05±2.03 | 14.00±9.78 | 18.74±7.10 | 26.41±10.97 | 43.30±1.68 |
| qEHVI-SF | 2 | 18.13±12.67 | 3.03±1.00 | 9.36±5.50 | 16.51±8.31 | 19.15±5.59 | 44.13±1.83 |
|  | 5 | **15.53±6.70** | **2.22±0.68** | **9.33±4.98** | **13.83±7.42** | **16.96±6.40** | 44.43±2.44 |
|  | 10 | 20.15±8.60 | 3.70±3.07 | 10.91±6.17 | 17.12±7.40 | 22.12±9.49 | **42.78±1.33** |

Table 3: Comparison of extreme Pareto capturing with respect to Maximum Spread (MS)↑

| Acquisition strategies | Batch size | Bi-1 | Bi-2 | Bi-3 | Tri-1 | Tri-2 | All |
|---|---|---|---|---|---|---|---|
| qEHVI | 2 | 876.16±112.74 | 67.63±8.25 | 431.45±170.08 | 956.63±166.71 | 525.12±114.95 | 968.89±88.40 |
|  | 5 | 944.89±186.81 | 70.58±8.92 | 465.05±155.17 | 1010.37±173.56 | 512.09±100.24 | 971.98±138.67 |
|  | 10 | 953.59±135.00 | 74.32±8.94 | 515.05±127.36 | 1055.08±162.75 | 460.63±129.17 | 1006.22±144.28 |
| QSVGD | 2 | 911.52±108.11 | 71.09±8.97 | 427.48±153.29 | 1075.84±237.17 | 483.38±92.83 | 984.77±147.88 |
|  | 5 | 994.27±167.27 | 71.37±7.05 | 467.78±126.22 | 1047.58±145.78 | 455.63±120.94 | 1008.11±137.09 |
|  | 10 | 967.97±144.84 | 67.80±9.68 | 454.77±154.16 | 1087.36±145.54 | 459.50±146.25 | **1032.97±147.26** |
| qEHVI-SF | 2 | 1025.78±161.75 | 73.51±5.81 | **524.49±129.78** | 1095.15±122.34 | **554.92±102.30** | 984.04±122.55 |
|  | 5 | **1145.63±179.89** | **76.83±3.73** | 499.61±112.38 | **1160.04±270.84** | 533.41±111.44 | 1002.14±152.16 |
|  | 10 | 974.65±109.28 | 70.85±10.07 | 480.01±128.35 | 953.47±146.65 | 478.82±144.99 | 1005.94±147.49 |

Table 4: Comparison of Pareto front uniformity with respect to Spacing (SP)↓

| Acquisition strategies | Batch size | Bi-1 | Bi-2 | Bi-3 | Tri-1 | Tri-2 | All |
|---|---|---|---|---|---|---|---|
| qEHVI | 2 | 1.19±0.09 | 1.35±0.13 | 1.42±0.10 | 1.38±0.09 | 1.12±0.21 | 0.75±0.04 |
|  | 5 | 1.08±0.10 | 1.13±0.11 | 1.33±0.09 | 1.25±0.10 | 0.99±0.18 | 0.77±0.07 |
|  | 10 | 0.99±0.08 | 1.04±0.10 | 1.28±0.10 | 1.11±0.08 | 0.74±0.14 | 0.67±0.04 |
| QSVGD | 2 | 1.12±0.11 | 1.27±0.10 | 1.36±0.09 | 1.28±0.10 | 1.05±0.13 | 0.75±0.03 |
|  | 5 | 1.14±0.07 | 1.18±0.10 | 1.33±0.08 | 1.24±0.08 | 0.85±0.18 | 0.73±0.05 |
|  | 10 | 1.03±0.09 | 1.05±0.09 | 1.24±0.12 | 1.09±0.07 | 0.73±0.08 | 0.68±0.04 |
| qEHVI-SF | 2 | 0.76±0.06 | 1.06±0.18 | 1.25±0.12 | 1.14±0.10 | 1.18±0.14 | 0.69±0.05 |
|  | 5 | 0.70±0.05 | 0.87±0.11 | 1.16±0.12 | 0.94±0.07 | 0.85±0.06 | 0.68±0.05 |
|  | 10 | **0.65±0.05** | **0.73±0.10** | **1.10±0.12** | **0.78±0.10** | **0.67±0.10** | **0.64±0.05** |

For performance comparison by IGD, shown in Table 2, and Spacing (SP), shown in Table 4, qEHVI-SF demonstrates superior performance compared to other acquisition strategies. This indicates that, relative to qEHVI and QSVGD, qEHVI-SF not only achieves better coverage of the objective space but also produces a more uniformly distributed Pareto front. Regarding the Maximum Spread metric shown in Table 3, QSVGD outperforms qEHVI-SF on case `all` composition, suggesting that QSVGD is occasionally able to capture more extreme solutions by providing candidates with greater diversity. This observation is consistent with the QSVGD EMD performance presented in Figure A.5f. Performance comparison results with the three metrics, IGD, Maximum Spread (MS), and Spacing (SP), provide strong complementary evidence that qEHVI-SF recovers the Pareto front more effectively and reinforce the conclusion drawn from the reported results on the rediscovery rate in the main text.

To clearly illustrate the differences between acquisition strategies: qEHVI, QSVGD, and qEHVI-SF, we select the bi-objective tasks with the batch size of two to visualize and compare the search behavior of these methods.

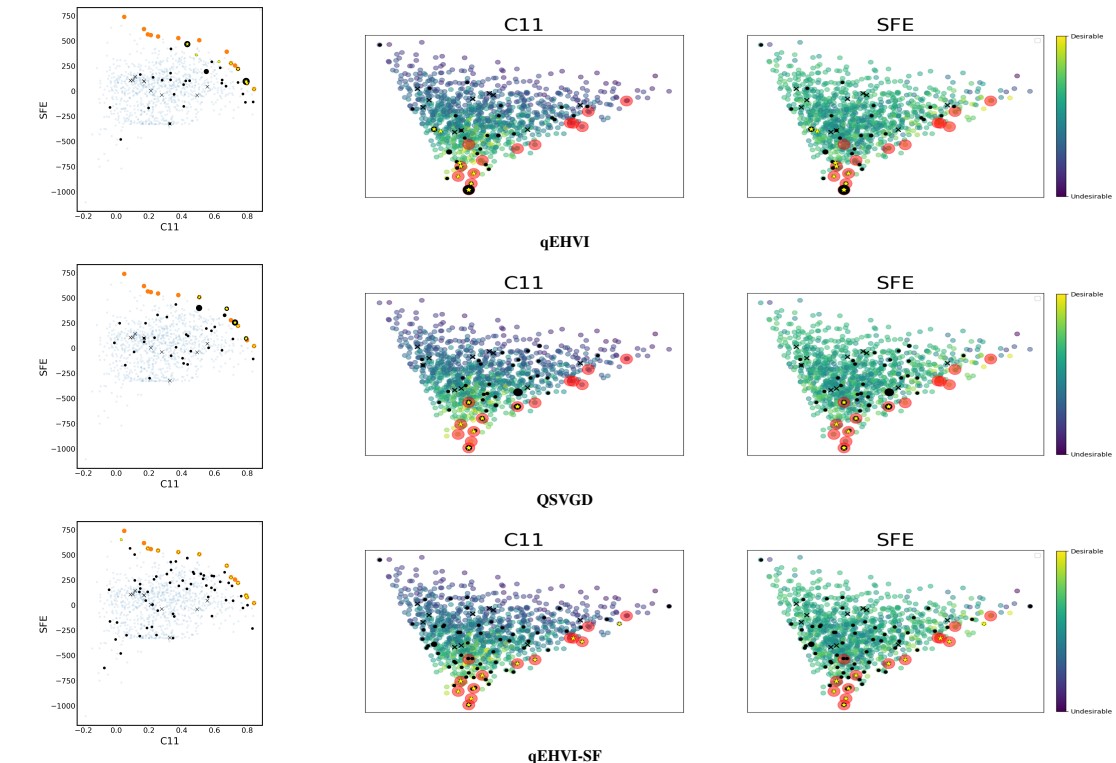

Figure A.6: Comparison of search behaviors for three acquisition strategies: qEHVI, QSVGD, and qEHVI-SF on case `Bi-1`. The first column shows the search trajectories in the objective space. The remaining two columns illustrate the distribution of Pareto optimal solutions in the design space. Background: feasible region; Red points: Pareto optimal solutions; Black dots: acquired candidates, with radius proportional to the time of selections; Yellow stars: optimal solutions among the acquired candidates; Black crosses: initial training points.

From Figures A.6, A.7, and A.8, we observe that qEHVI-SF consistently achieves better Pareto front coverage than qEHVI and higher solution quality than QSVGD. This improved rediscovery performance is primarily due to its lower risk of oversampling. The significantly lower number of large radius black dots in these plots visualizing qEHVI-SF's results suggests that our qEHVI-SF avoids repeatedly selecting the same candidates, unlike qEHVI and QSVGD.

Interestingly, although QSVGD is designed to promote diversity and reduce oversampling, it still exhibits redundant selections in all three cases. This is likely because QSVGD enforces diversity only within each batch, without accounting for previously evaluated points. While QSVGD can sometimes enhance diversity, as seen in Figure A.8. However, a closer look at the upper left region reveals that it misses some high-quality Pareto optimal solutions. In contrast, qEHVI-SF successfully identifies and selects these high-quality candidates.

## A.4 USE OF LARGE LANGUAGE MODELS

Large Language Models are only used to check vocabulary and grammar for polishing purpose.

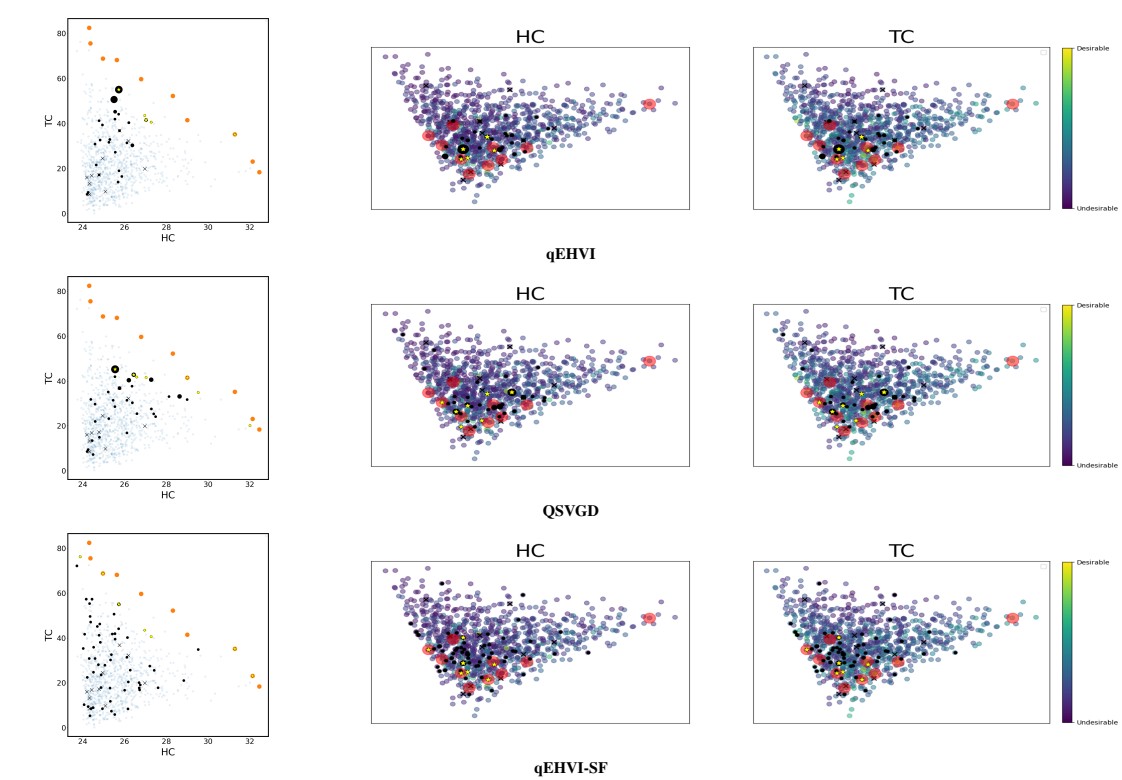

Figure A.7: Comparison of search behaviors across three acquisition strategies for case `Bi-2`.

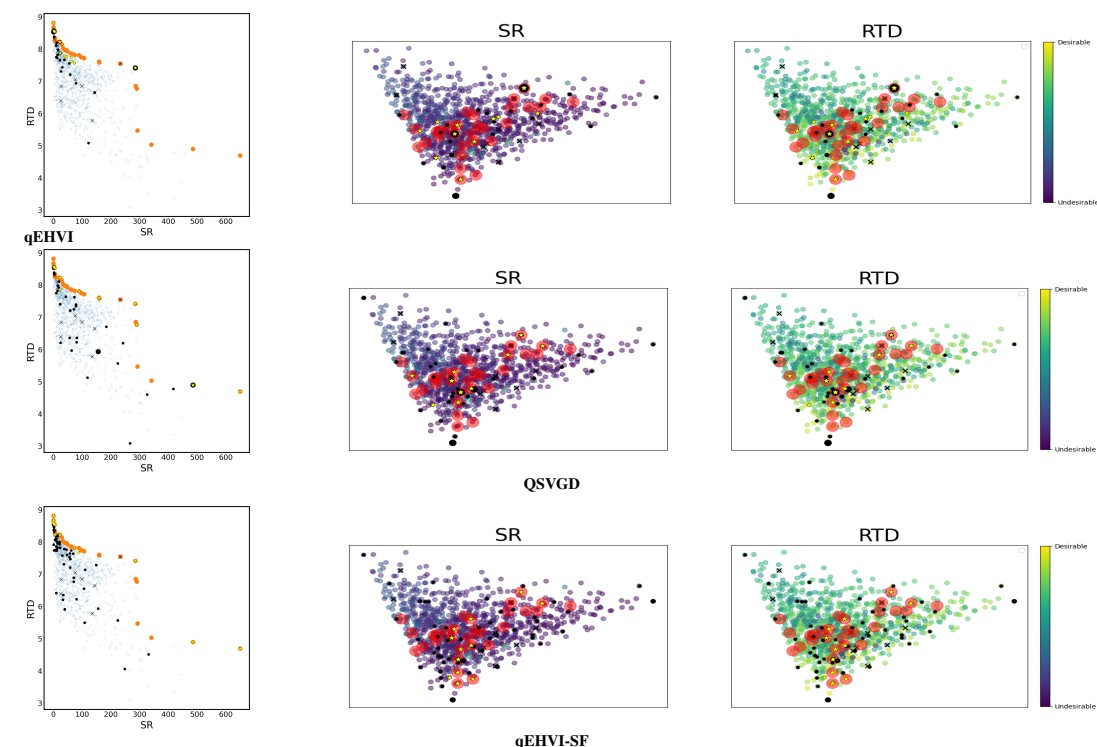

Figure A.8: Comparison of search behaviors across three acquisition strategies for case `Bi-3`.

