# OpenReview forum: "Probability of Matching for Pareto Coverage"
_ICLR.cc/2026/Conference — Submitted to ICLR 2026_

### Official Review · Reviewer_kUdU · 2025-10-24

**Soundness:** 2
**Presentation:** 3
**Contribution:** 2
**Rating:** 4
**Confidence:** 3

**Summary:**

This paper proposes a new acquisition strategy for expensive multi-objective Bayesian optimization (MOBO) that jointly promotes the quality and diversity of solutions in the Pareto set. The key idea is to estimate the probability of matching, which encompasses (i) the probability that batch points are Pareto-optimal, and (ii) the probability that they collectively cover the entire Pareto front. The authors approximate these terms using qEHVI for quality and a space-filling design for coverage, yielding an acquisition function that balances exploration and exploitation without additional hyperparameters.

**Strengths:**

- Writing is good, easy to follow.
- The paper clearly identifies the limitation of existing MOBO approaches that overemphasize quality while neglecting diversity. By modeling diversity in the design space rather than the objective space, the method avoids surrogate bias and is potentially more robust for problems with dispersed Pareto fronts.
- The probability of matching $P(X = {X}^{\star})$  is an intuitive and compelling way to frame the batch MOBO problem. Factorizing it into a quality term $P(X \subseteq X^\star)$ and coverage term $P({X}^\star \subseteq X \mid X \subseteq {X}^\star)$ is a clean, logical decomposition.
- The introduction of the Expected Minimum Distance (EMD) metric is a useful addition. Measuring coverage in the design space offers a stricter and more informative alternative to traditional objective-space metrics such as IGD.

**Weaknesses:**

- The theoretical formulation defines the coverage probability $P({X}^\star \subseteq X | X \subseteq {X}^\star)$, yet the practical implementation replaces this with an approximation $P({X}^\star \subseteq A^r_X | X \subseteq {X}^\star)$, where $A^r_X$ is a union of balls. The final acquisition (Eq. 8) relies on a minimum-distance heuristic $\min(\Delta(X, {X}), \Delta(X, X_n))$, but the paper does not provide a rigorous justification linking this heuristic to maximizing the coverage probability. Could the authors provide a more formal bridge between the theoretical coverage formulation and the adopted minimum-distance heuristic?
- The coverage probability is approximated using minimax distance, which may not adequately capture complex or multimodal Pareto structures. Could alternatives like kernel-based diversity be explored?
- This paper focuses on the MOBO problems that have many optimal solution regions, aiming to demonstrate the effectiveness of the proposed coverage term. However, the results do not significantly outperform the baselines; results on ZDT/DTLZ (Appendix) show smaller gains. The author should provide more diverse problems to strengthen claims.
- The authors mentioned that a larger batch size leads to higher $P(X^\star \subseteq X)$ while it may reduce $P(X \subseteq X^\star)$. However, the result indicates that each batch size setting illustrates similar performance. There's no clear guidance on how to choose a batch size in practice for new problems.  Could you conduct more experiments for this phenomenon?

**Minor issues:**
- Some figures could better label axes or explain color scales for clarity.
- The paper should be presented in the template of ICLR 2026, not ICLR 2025.

**Questions:**

Kindly address weaknesses.

---

### Official Review · Reviewer_EaKA · 2025-10-29

**Soundness:** 2
**Presentation:** 3
**Contribution:** 2
**Rating:** 2
**Confidence:** 4

**Summary:**

This paper proposes a new multi-objective acquisition function inspired by maximizing the probability that sampled batch points match the Pareto frontier. They use set theory identities to decompose this probability into two parts one representing the probability of sampled batch being Pareto optimal, and the other representing how well the batch X covers the entire Pareto set. Since these probabilities are intractable to compute directly, they approximate the first using qEHVI as a heuristic proxy for solution quality, and the second maximizes the minimum pairwise distance between batch points and previously sampled points, under the assumption that well-spaced points are more likely to cover dispersed Pareto optimal regions. The algorithm is benchmarked on synthetic problems including Gaussian mixture and RE4-7-1, standard MOBO benchmarks like DTLZ and ZDT families, and real world problems, demonstrating consistently better performance across hypervolume, inverted generational distance, and a new design-space coverage metric called expected minimum distance, while maintaining computational efficiency comparable to standard qEHVI.

**Strengths:**

- The set-theoretic decomposition is conceptually elegant and refreshing, providing a principled justification for the hyperparameter-free multiplicative combination of quality and diversity terms.
- The algorithm demonstrates competitive empirical performance across synthetic benchmarks and real-world materials discovery tasks.

**Weaknesses:**

- I think the main issue is the heuristic foundation of this paper, there is no theoretical justifying thet $p(X \subseteq X^*)$ can be approximate by qEHVI and in fact, I don't think this ever hold and one can justify by using exhaustive MC sampling as approximation.
- Similar concerns apply to the coverage term: the connection between space-filling (minimum distance) and coverage probability is purely heuristic
- (Minor) Tables would benefit from bold text to highlight best performance for easier comparison.

**Questions:**

The acquisition function multiplies quality and coverage terms. As batch size $q$ increases, the quality term $P(X \subseteq X^*)$ may decrease significantly since it requires all $q$ points to simultaneously be Pareto optimal. While the outputs are correlated under the GP prior, does this mean the quality term diminishes with $q$ while the coverage term grows with $q$? If so, how does their product balance these opposing trends, and does this create a systematic bias toward smaller or larger batch sizes?

---

### Official Review · Reviewer_y2j9 · 2025-10-30

**Soundness:** 1
**Presentation:** 3
**Contribution:** 1
**Rating:** 2
**Confidence:** 4

**Summary:**

This paper proposes a new batch multi-objective Bayesian optimization acquisition strategy. It extends the popular qEHVI framework by incorporating the proposed Probability of Matching in order to trade-off convergence and diversity. An empirical comparison is performed on synthetic benchmarks and real-world tasks.

**Strengths:**

The paper proposes a novel acquisition strategy that jointly considers both convergence and diversity in batch multi-objective Bayesian optimization - a promising research topic in Bayesian optimization.

**Weaknesses:**

1) Maximizing HV is actually also diversifying solutions as the true Pareto front always has the maximum HV value. It seems to me that this plays a similar role with maximising the coverage of solutions.

2) During HV-related search, the reference point may not be a big issue as it can be set adaptively based on the nondominated solutions found.

3) The metric used for maximizing coverage is like the well-known uniformity metric in multi-objective evolutionary optimization, Spacing (SP), which measures the maximim distance between two nearest solutions.

4) IGD is not first proposed in Ishibuchi et al 2025. Please cite the orignal paper.

5) Not quite understand why you consider the "IGD" in the decision space. You didn't consider the diversity of solutions in the decision space since your method cares about the diversity in the objective space. There could be a situation that there is perfectly distributed solution set in the decision space, but poorly distributed mappings in the objective space - I think such a case can't verify your algorithm?

6) Not quite understand why your algorithm obtains a better HV compared to qEHVI since the latter directly optimize HV, but you try to strike a balance between HV and SP. Did you use the same reference setting strategy? If yes, what is it?

7)  The proposed method is based on qEHVI and jointly estimates a batch of solutions, which may scale up poorly with the batch size [1] and perform worse than a sequential strategy [2,3]. When using a sequential acquisition strategy, qEHVI [2,3] can naturally produce a batch of solutions with good diversity and convergence through its iterative uncertainty updates. It would be useful to include a theoretical and empirical comparison between the proposed method and the sequential qEHVI.

8) The related work section is not very comprehensive with respect to MOBO methods. The experimental evaluation lacks comparisons with several well-established MOBO baselines, such as Sobol sampling, ParEGO, and Joint Entropy Search (JES) [4].


[1] Daxberger, E. A.; and Low, B. K. H. 2017. Distributed batch Gaussian process optimization. In Proceedings of the 34th International Conference on Machine Learning, 951–960. PMLR.

[2] Samuel Daulton, Maximilian Balandat, and Eytan Bakshy. Differentiable expected hypervolume improvement for parallel multi-objective Bayesian optimization. In Advances in Neural Information Processing Systems, volume 33, pages 9851–9864. Curran Associates, Inc., 2020.

[3] Samuel Daulton, Maximilian Balandat, and Eytan Bakshy. Parallel Bayesian optimization of multiple noisy objectives with expected hypervolume improvement. In Advances in Neural Information Processing Systems, volume 34. Curran Associates, Inc., 2021.

[4] Ben Tu, Axel Gandy, Nikolas Kantas, and Behrang Shafei. Joint entropy search for multi-objective Bayesian optimization. Advances in Neural Information Processing Systems, 35:9922–9938, 2022.

**Questions:**

Could you please respond to my comments 1, 3, 5, 6, 7 in the Weaknesses section?

---

### Official Review · Reviewer_jkQL · 2025-11-01

**Soundness:** 3
**Presentation:** 2
**Contribution:** 3
**Rating:** 4
**Confidence:** 4

**Summary:**

This paper proposes qEHVI-SF, a novel batch multi-objective Bayesian optimization (MOBO) algorithm that introduces the Probability of Matching (PoM) — a probabilistic measure quantifying how likely a batch of candidate points matches the true Pareto optimal set. The approach factorizes this probability into two components: (i) the probability that all batch points are Pareto optimal, and (ii) the probability that they collectively cover the Pareto front. By combining qEHVI for quality estimation with a space-filling principle for diversity, yielding an integrated acquisition function that promotes both optimality and coverage. Extensive experiments on synthetic benchmarks and a real-world materials design case study show consistent improvements in hypervolume, IGD, EMD, and rediscovery ratio, while maintaining comparable computational efficiency to qEHVI. The paper is well-motivated, technically sound, and experimentally strong, though probabilistic factorization lacks rigorous justification.

**Strengths:**

1. Unified acquisition design: Integrates performance and diversity without manual hyperparameter tuning.
2. Comprehensive experiments: Covers both synthetic and real-world alloy design tasks, showing consistent gains.
3. Introduces a useful coverage metric (EMD) in design space, adding depth to MOBO evaluation.
4. Maintains computational efficiency comparable to qEHVI, demonstrating practical scalability.

**Weaknesses:**

1. The probabilistic factorization (Eq. 7–8) is intuitive but lacks formal theoretical proof.
2. Ablation study for the space-filling component is missing, making it unclear how much it contributes individually.
3. Some figures and equations are overly dense, affecting clarity.
4. No discussion on extending the framework to constrained or preference-based MOBO.
5. Theoretical limitations section (Section 5) could be expanded to strengthen formal grounding.

**Questions:**

1. Can the authors provide a more formal justification for the decomposition of Eq. (7–8)?
2. How sensitive is the method to batch size q? Could adaptive q be explored?
3. How does qEHVI-SF behave when the Pareto front is discontinuous or highly non-convex?
4. Could the approach be extended to constrained or preference-aware MOBO scenarios?
5. Would the authors consider releasing implementation code to facilitate reproducibility?

---

### Meta-Review · Area_Chair_FJYy · 2025-12-06

**Summary:**

This work introduces a new acquisition function for MOBO called Probability of Matching, which is a modification of the well-known qEHVI family of acquisition functions to consider space-filling strategies.

The reviewers mostly agree the empirical work is comprehensive in this paper, and performance is decent.

However, there are a number of key concerns that remain unaddressed:

- the method lacks formal rigour and justification
- some key ideas and theoretical comparisons to concepts from the MOBO and MOEA literature are absent in the related work section
- some key ablations are missing
- key baselines comparisons are missing

Thus I feel compelled to suggest this paper is not ready for acceptance.

**Reviewer Concerns:**

No concerns addressed

**Reviewer Scores:**

No score changes

---

### Decision · Program_Chairs · 2026-01-26

Reject